# Enhancing the Yield, Quality and Antioxidant Content of Lettuce through Innovative and Eco-Friendly Biofertilizer Practices in Hydroponics

Hayriye Yildiz Dasgan [1,*] , Dilek Yilmaz [1], Kamran Zikaria [1], Boran Ikiz [1] and Nazim S. Gruda [2]

1 Department of Horticulture, Agricultural Faculty, Cukurova University, 01330 Adana, Turkey; dlk_ylmz01@hotmail.com (D.Y.); muhammadkamran7253@gmail.com (K.Z.); bikiz@cu.edu.tr (B.I.)
2 Institute of Plant Sciences and Resource Conservation, Department of Horticultural Sciences, University of Bonn, D-53113 Bonn, Germany; ngruda@uni-bonn.de
* Correspondence: dasgan@cu.edu.tr

**Abstract:** Hydroponics is a contemporary agricultural system providing precise control over growing conditions, potentially enhancing productivity. Biofertilizers are environmentally friendly, next-generation fertilizers that augment product yield and quality in hydroponic cultivation. In this study, we investigated the effect of three bio-fertilizers in a hydroponic floating system, microalgae, plant growth-promoting rhizobacteria (PGPR) and arbuscular mycorrhizal fungi (AMF), combined with a 50% reduction in mineral fertilizer, on lettuce yield and quality parameters including antioxidants: vitamin C, total phenols and flavonoids. The treatments tested were: 100% mineral fertilizer (control 1), 50% mineral fertilizer (control 2), 50% mineral fertilizer with microalgae, 50% mineral fertilizer with PGPR and 50% mineral fertilizer with AMF. The research was conducted during the winter months within a controlled environment of a glasshouse in a Mediterranean climate. The PGPR comprised three distinct bacterial strains, while the AMF comprised nine different mycorrhizal species. The microalgae consisted of only a single species, Chlorella vulgaris. AMF inoculation occurred once during seed sowing, while the introduction of PGPR and microalgae occurred at 10-day intervals into the root medium. Our findings revealed that the treatment with PGPR resulted in the highest growth parameters, including the lettuce circumference, stem diameter and fresh leaf weight. The 100% mineral fertilizer and PGPR treatments also yielded the highest lettuce production. Meanwhile, the treatment with AMF showed the highest total phenol and flavonoid content, which was statistically similar to that of the PGPR treatment. Furthermore, the PGPR recorded the maximum range of essential nutrients, including nitrogen (N), potassium (K), iron (Fe), zinc (Zn) and copper (Cu). Thus, the inclusion of PGPR holds promise for optimizing the lettuce growth and nutrient content in hydroponic systems. In conclusion, PGPR has the potential to enhance nutrient availability in a floating hydroponic system, reducing the dependence on chemical fertilizers. This mitigates environmental pollution and fosters sustainable agriculture.

**Keywords:** *Lactuca sativa* L.; plant growth promoting rhizobacteria; mycorrhiza; microalgae; floating culture; soilless culture; sustainable agriculture





## 1. Introduction

Lettuce (*Lactuca sativa* L.) is a leafy vegetable of great importance belonging to the *Asteraceae* family. It is a rich source of vitamins such as A, C, folate and K, which help boost the immune system and maintain bone health. Antioxidants, including flavonoids, phenolic acids and carotenoids, protect the body against free radicals that can cause cell damage and diseases [1]. Lettuce leaves are typically consumed fresh in salads or minimally processed products, such as fresh-cut products, mixed salads and baby products. It is in demand year-round by consumers in the market [2]. Lettuce is relatively easy to cultivate

and has a short growing cycle, making it suitable for year-round hydroponic cultivation. Lettuce is a leading leafy vegetable grown hydroponically [3].

The continuous rise in the world's population, decreasing agricultural lands, rising urbanization, climate change and other stress factors lead to a loss in yields and reduced production [4]. These challenges in agriculture, in turn, pose new hurdles for global food safety and nutrition [5]. The changing weather and climate conditions have exacerbated the issue of water scarcity [3]. Agriculture is the largest sector that consumes water globally, accounting for approximately 70% of the total water demand [6,7]. Implementing an integrated irrigation and fertilization program is essential to achieving a maximum yield and high-quality production while adhering to an eco-friendly approach [8,9]. There is a growing interest in innovative farming practices characterized by a high water use efficiency and a high yield per unit area, driven by novel plant nutrition techniques.

Adopting new technologies has increased humanity's ability to address the challenges of limited resources. Hydroponics is considered an alternative to traditional agricultural systems [3,10]. Hydroponics is a soilless cultivation system widely adopted for advanced crop production worldwide [11–13]. In this farming system, plants are nourished with a combination of micro and macro nutrients in the water, which benefits achieving high crop production [14,15]. Cultivating plants, such as vegetables, herbs and flowers, by supplying a nutrient solution with optimal nutrient concentrations is essential for plant growth [16,17]. Implementing this advanced farming technique does not require soil for raising crops [18]. Greenhouse hydroponics is a sustainable agricultural system with controlled conditions that enhance plant health and growth [19,20]. Hydroponic systems help conserve water and provide favorable environmental conditions for vegetables.

Floating hydroponic culture is a closed system representing one of the most water-efficient systems among existing hydroponic cultivation methods. One of its pivotal characteristics is the utilization of a substantial water volume, which serves as a reservoir for fertilization, oxygenation, water temperature regulation and cost-effective plant transportation through flotation [21]. This generous water buffer provides security and convenience unmatched by other cultivation systems [22]. Floating culture boasts several advantages, including yielding several crop cycles annually. This system optimizes space utilization and facilitates rapid turnover while eliminating the need for herbicides or fungicides. One of the significant benefits of floating culture is the maximization of nutrient solution usage with zero waste, rendering it an environmentally friendly approach [23,24]. This system requires a limited area, little labor, little time and low energy [21].

The excessive use of mineral–chemical fertilizers to achieve high yields per unit area has a detrimental impact on the environment. Therefore, researchers are striving to promote environmentally friendly organic fertilizers, such as biofertilizers, to reduce the environmental hazards caused by mineral fertilizers [8,25]. Biofertilizers were introduced as a sustainable alternative to chemical fertilizers, posing adverse effects on living organisms and the environment. Furthermore, biofertilizers have proven highly efficient, positively affecting the germination, growth, yield and crop quality [26,27]. These environmentally friendly fertilizers also contribute to reducing carbon emissions in the environment. The practical application of rhizobacterial processes, such as biofertilization, phyto-remediation and cross-protection, depends on the ability to produce desired strains and the existing population of microorganisms [28]. Many rhizobacteria benefit plant growth and enhance plant tolerance to biotic and abiotic stresses [29,30].

Naturally, bio-fertilizers consist of various strains of fungi and bacteria, intending to reduce the use of chemical fertilizers in various applications [31]. Bio-fertilizers enhance plant growth, including increasing the number of leaves, leaf area, length, shoot fresh and dry weight and root fresh and dry weight [32–34]. Additionally, bio-fertilizers can mobilize mineral elements from an unavailable to an available form, thereby increasing the availability of both micro and macro nutrients for plants [8,35]. Soil naturally contains beneficial microorganisms supporting plant nutrition, producing phytohormones, controlling phytopathogens and improving the soil structure. However, soilless cultivation systems

often lack these beneficial microorganisms [36,37]. For this reason, in this study, living microorganisms were integrated into hydroponic lettuce cultivation.

Several PGPR have been investigated within hydroponic systems for potential applications in agriculture, serving as biofertilizers, biocontrol agents and bioremediators [38]. However, a study for lettuce involving the reduction in mineral fertilizers in a recycling floating culture system in favor of using living biofertilizers has not been conducted previously. The objective of this study was to explore the utilization of biofertilizers, including PGPR, AMF and microalgae, to enhance lettuce nutrition through environmentally friendly crop management practices and reduce the reliance on mineral–chemical fertilizers.

## 2. Materials and Methods

### 2.1. Plant Material

This experiment was conducted during the winter season of 2019 in a glass greenhouse at the University of Cukurova, Adana, Turkey (36°59′ N, 35°18′ E, 20 m above sea level) (Figure 1). The vegetable material used in the experiment was *cv.* 'Dragone' is a Batavia type of green lettuce provided by the Vilmorin seed company. The 'Dragone' lettuce used in the experiment is a mid-early variety with a harvest time of 55–60 days under suitable temperature conditions when grown in soil. It is known for its high attractiveness and resistance to downy mildew and aphids.

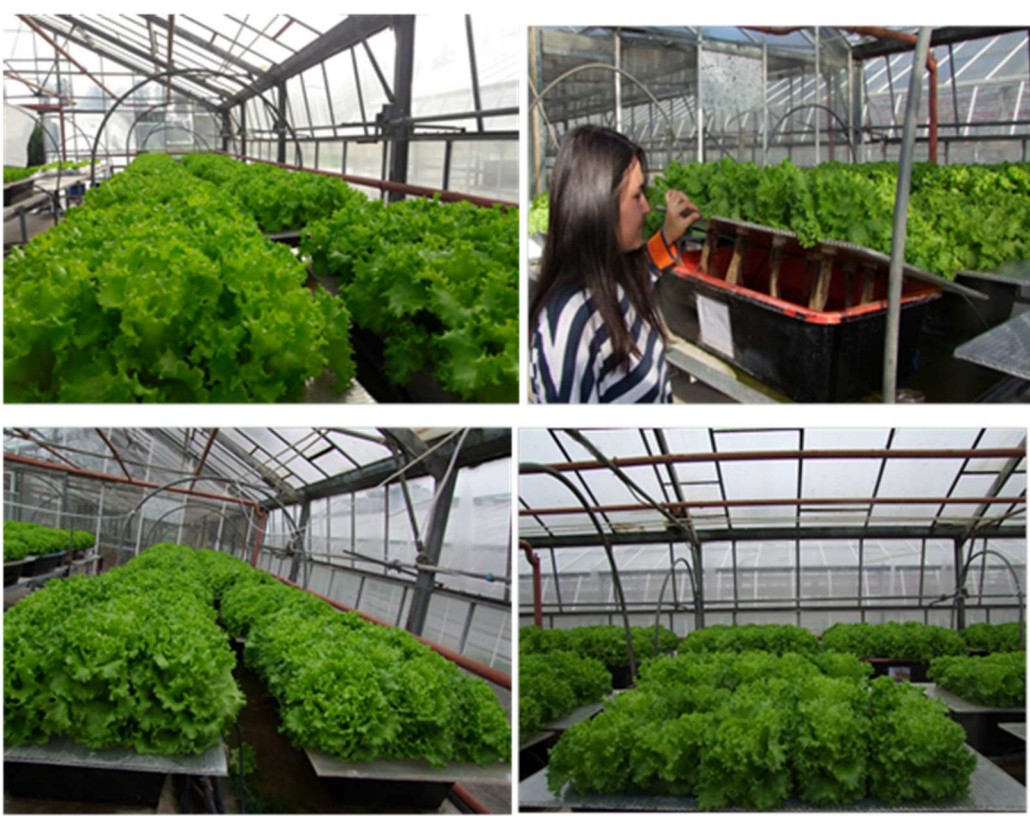

**Figure 1.** Lettuce plants treated with biofertilizers in floating hydroponic culture.

### 2.2. Bioertilizers Used in the Experiment

The experiment consisted of three bio-fertilizers inoculated into the root area:

Microalgae (*Chlorella vulgaris*): This microalgae, produced at Cukurova University, was diluted 40 times before inoculation. A total of 25 mL of microalgae per liter was added from a culture concentration of $2 \times 10^7$ (colony units per milliliter) at the plant root level [39,40]. PGPR: A mixture of three species, including *Bacillus subtilis*, *Bacillus megagaterium* and *Pseudomonas fluorescens*, was obtained from NGB (The Next-Generation Biotechnology) under the trade name 'Rhizofill.' A quantity of 50 mL of bacteria with a concentration of

$1 \times 10^9$ colony units per milliliter was inoculated into a 50-liter nutrient solution tank (1 mL per liter) every 10 days [40].

AMF: These fungi were obtained from the Bioglobal company under the trade name 'ERS' (Endo Root Soluble) and contained species like *Glomus intraradices*, *Glomus aggregatum*, *Glomus mosseae*, *Glomus clarum*, *Glomus monosporus*, *Glomus deserticola*, *Glomus brasilianum*, *Glomus etunicatum* and *Gigaspora margarita*, with a concentration of $1 \times 10^4$ colony units per gram. Lettuce seeds were inoculated with AMF during sowing, with 1000 spores per plant [39,40]. The treatments involved the substitution of bio-fertilizers for reduced mineral fertilizers. The study comprised five treatments, described as follows:

T1: 100% mineral fertilization (MF) control 1
T2: 50% mineral fertilization control 2
T3: 50% mineral fertilization + Microalgae
T4: 50% mineral fertilization + PGPR
T5: 50% mineral fertilization + AMF

### 2.3. Plant Growing Conditions

When sowing, the lettuce seeds were inoculated with AMF for T5, and PGPR were added to the nutrient solution of the floating culture every 10 days during cultivation. After 38 days from sowing, at the seedling stage with five true leaves, the seedlings were transferred to the floating culture system in the greenhouse. The climatic conditions inside the glasshouse ranged from 20–23 °C during the day to 13–15 °C at night, with relative humidity between 50 and 60%, and exposure to natural sunlight conditions. In this experiment, the plants were grown in randomized blocks with a plant spacing of 15 cm × 15 cm, resulting in a density of 44.44 plants per square meter. The experiment was designed with four replications, and each replication consisted of 15 plants (60 plants per treatment). Hydroponic culture nutrients were reduced by 50% when transferring the seedlings and applying the biofertilizers. Each plastic container with a volume of 50 liters allowed the roots to be completely submerged in the nutrient solution and was well-aerated to maintain dissolved oxygen in the water (Figure 1). The 'Floating Water Culture' method was used for growing the lettuce plants.

### 2.4. Plant Nutrition

Two nutrient stock solution tanks were utilized, as outlined in Table 1, with the concentrations of nutrient elements for lettuce provided in Table 2 [41]. The pH of the solution was checked and maintained daily, falling within the range of 5.7 to 6.0. The nutrient solution's electrical conductivity (EC) values gradually increased with plant growth, reaching 1.5, 2.0 and 2.4 dS m$^{-1}$ in the 100% mineral fertilizer control application. In the bio-fertilizer applications, the use of mineral fertilizers was reduced by 50% compared to the 100% mineral fertilizer control application. The lettuce plants were grown in the floating hydroponic system for 40 days (Figure 2). The nutrient solution was renewed every ten days.

**Table 1.** Mineral fertilizers used in the experiment and the way they are included in the stock solutions in the feeding system.

| Stock A | Stock B |
|---|---|
| Calcium nitrate | Potassium sulfate |
| Fe—EDDHA | Mono potassium phosphate |
| | Magnesium sulfate |
| | Potassium nitrate |
| | Microelements |
| | Zinc sulfate |
| | Boric acid |
| | Manganese sulfate |
| | Copper sulfate |
| | Ammonium molybdate |

**Table 2.** Concentration ranges of nutrients used in the 100% MF control application in the experiment.

| Element | mg L$^{-1}$ |
|---|---|
| N | 150–220 |
| P | 30–40 |
| K | 270–312 |
| Ca | 170–210 |
| Mg | 50–65 |
| Fe | 3.00–5.00 |
| Zn | 0.30–0.55 |
| B | 0.70–0.97 |
| Cu | 0.20–0.30 |
| Mo | 0.10–0.20 |
| Mn | 0.55–0.96 |

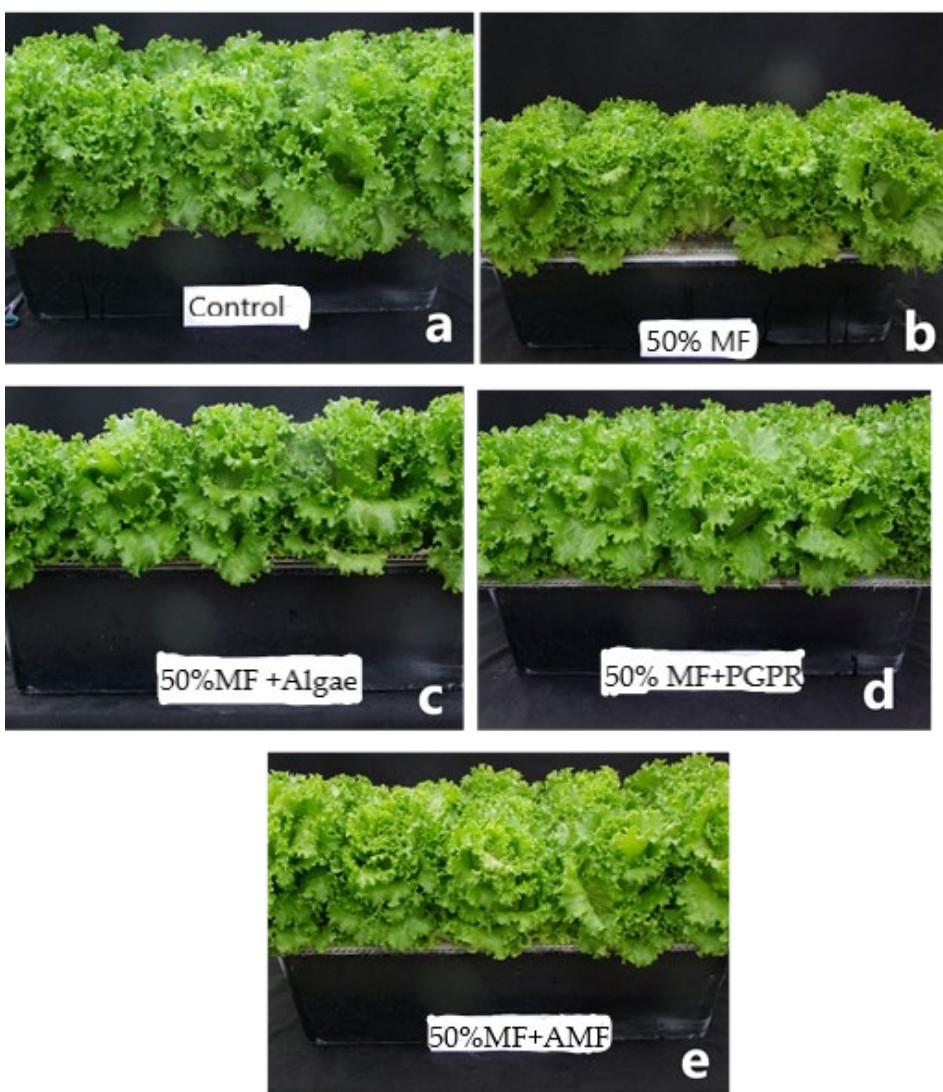

**Figure 2.** The impact of different biofertilizers at the harvest stage on the lateral profile of lettuces in cultivation tanks: (**a**) 100% mineral fertilization, control 1; (**b**) 50% mineral fertilization, control 2; (**c**) 50% mineral fertilization + Microalgae; (**d**) 50% mineral fertilization + PGPR; (**e**) 50% mineral fertilization + AMF.

### 2.5. Lettuce Harvest

Lettuce plants were harvested 40 days after transplanting, and the total yield values for each replication, as well as the measurements of the green parts of the plants, were recorded individually.

### 2.6. Measurements of Plant Growth Parameters

At harvest time, the yield per unit area (kg m$^{-2}$) was calculated by summing the total weight of 15 plants in each replication, and leaf counts were recorded. Additionally, various measurements were taken, including the height, plant circumference, plant width, root length and primary stem diameter of the lettuce plants.

A digital penetrometer (Bareiss HPE-III-Fff, ABQ Industrial, Rolling Meadows, IL, USA) was employed to measure lettuce firmness, recorded in kilograms, from the outer part of the leaves. A leaf area meter (Li-3100, LICOR, Lincoln, NE, USA) was used to determine the leaf area per plant in square centimeters. Following the harvest, plants were weighed on a digital balance to ascertain the fresh leaf weight per plant in grams. An SPAD–chlorophyll meter (Minolta 502, Osaka, Japan) was utilized to measure the chlorophyll content in the leaves. The luminosity (L) and chromaticity (a (red-green axis) and b (blue-yellow axis)) values were digitally displayed on a portable digital handheld color spectrophotometer device (HunterLab, Reston, VA, USA) for the harvested lettuce leaves. Fresh leaves were subsequently dried at 65 °C for 48 h, and the dry weight per plant was determined.

### 2.7. Measurements of Total Soluble Solids (TSS), pH and Electrical Conductivity (EC) in Lettuce Leaves

The juice from the lettuce leaves was utilized to measure the TSS (Total Soluble Solids), pH and EC (Electrical Conductivity) using a digital refractometer (Atago PR-101, Tokyo, Japan) and a pH and EC meter (WTW pH/Cond 3320, Weilheim, Germany).

### 2.8. Measurement of Nitrate Concentration

The nitrate concentration in the lettuce leaves was determined using the salicylic acid method described by Cataldo et al. [42] and measured calorimetrically at 410 nm. The concentrations were expressed as milligrams per kilogram of fresh weight.

### 2.9. Measurement of Ascorbic Acid Content (Vitamin C)

Vitamin C was determined using the modified method of Elgailani et al. [43]. Lettuce leaves were blended in a juicer machine for extraction. Five milliliters of the extract were added to 45 milliliters of 0.4% oxalic acid and then filtered. One milliliter from the filtered solution was added to 9 milliliters of 2,6-dichlorophenolindophenol and mixed. Finally, all the samples were read at 520 nm using a UV spectrophotometer.

### 2.10. Measurement of Total Phenolic and Flavonid Substances

Total phenolics were determined in lettuce by modifying the spectrophotometric method described by Spanos and Wrolstad [44]. The readings were calculated using the absorbances at a wavelength of 765 nm in a spectrophotometer (UV-1700 PharmoSpec Shimadzu, Kyoto, Japan), and the calibration curve was prepared with gallic acid. Total flavonoids were analyzed using a spectrophotometer at 415 nm, following the method by Quettier-Deleu et al. [45]. The total amount of flavonoid substances was calculated using the calibration curve based on rutin.

### 2.11. Leaf Mineral Nutrient Analysis

The leaf mineral nutrient analysis was conducted by taking one-quarter of five plants from each replication at harvesting, following the method described in Dasgan et al. [39]. The analysis covered macro (N, P, K, Mg, Ca) and micro (Fe, Mn, Cu, Zn) nutrients. Lettuce leaf samples were cleaned to prevent contamination, washed three times with distilled water and dried at 65 °C for 48 h using the oven dry method. The dried samples were

ground to a 40-mesh size using a leaf grinding machine. For the analysis of K, Ca, Mg, Na, Fe, Mn, Zn and Cu, 0.2 g of the ground samples was incinerated at 550 °C for 5 h, and the resulting ash was dissolved in 3.3% ($v/v$) HCl and then filtered. The prepared samples for K, Ca, Mg and Na were analyzed in the emission mode, while Fe, Mn, Zn and Cu were analyzed in the absorbance mode using an Atomic Absorption Spectrophotometer device (Varian FS220, Las Vegas, NV, USA). Leaf nitrogen and phosphorus were determined using the Kjeldahl and Barton methods, respectively, as described in [40].

### 2.12. Statistical Analyses

Four biological replicates were employed for each biofertilizer treatment, with each biological replicate comprising five technical replications. The analysis of variance (ANOVA) was performed using the JMP statistical program (Version 7.0, Statistical Software, 2007). The means of the treatments were compared with the least significant difference (LSD) test at the $p \leq 0.05$ level. Furthermore, all independent variables underwent principal component analysis (PCA) and multiple variable analyses utilizing the Pearson correlation matrix through the ClustVis software 2.0, accessible at https://biit.cs.ut.ee/clustvis/ (accessed on 11 October 2023).

## 3. Results and Discussion

### 3.1. Effects of Biofertilizers on Plant Growth Parameters

The plants that exhibited the most robust growth parameters, including the plant height, plant circumference, plant width, stem diameter and head firmness, were those treated with 50% MF + PGPR biofertilizer, closely followed by those subjected to the 50% MF + AMF biofertilizer application. In contrast, the 50% MF + Microalgae application resulted in the least favorable growth of the plants (Table 3). As PGPR and AMF result in crispier leaves, the cumulative sum of leaves also creates a firmer lettuce.

**Table 3.** Effects of the biofertilizers on lettuce plant growth parameters—I.

| Treatments | Lettuce Height (cm) | Lettuce Circumference (cm) | Lettuce Width (cm) | Stem Diameter (mm) | Lettuce Head Firmness (kg cm$^{-3}$) |
|---|---|---|---|---|---|
| 100% MF | 29.05 ± 0.87 [a] | 50.66 ± 1.8 [a] | 34.56 | 22.48 ± 0.82 [ab] | 0.603 ± 0.004 [b] |
| 50% MF | 26.81 ± 0.71 [b] | 47.81 ± 2.6 [ab] | 30.56 | 22.99 ± 1.52 [ab] | 0.477 ± 0.030 [c] |
| 50% MF + Microalgae | 25.86 ± 2.45 [b] | 44.40 ± 2.4 [b] | 29.10 | 20.19 ± 1.50 [b] | 0.611 ± 0.039 [ab] |
| 50% MF + PGPR | 29.35 ± 1.06 [a] | 51.76 ± 1.8 [a] | 32.10 | 24.64 ± 1.25 [a] | 0.654 ± 0.050 [a] |
| 50% MF + AMF | 27.30 ± 0.12 [ab] | 50.66 ± 1.4 [a] | 31.15 | 24.01 ± 3.26 [a] | 0.648 ± 0.034 [a] |
| LSD$_{0.05}$ | 2.110 | 4.75 | ns | 2.923 | 0.0497 |
| $p$-value | 0.0156 | 0.0337 | 0.3156 | 0.0500 | <0.0001 |

MF: Mineral fertilizer, PGPR: *Plant growth-promoting rhizobacteria*, AMF: Arbuscular mycorrhizal fungi, LSD: The least significant difference between the means ($p < 0.05$), ns: Non-significant. Four biological replicates were employed for each biofertilizer treatment, with each biological replicate comprising five technical replications. Different letters indicate significant differences.

They play a crucial role in enhancing plant growth through various mechanisms. They facilitate phosphorus dissolution, nitrogen fixation and an improved mineral uptake, thereby promoting efficient nutrient utilization and enhancing both shoot and root development [28,38,46]. Additionally, PGPR can enhance a plant's resistance to diseases and abiotic stresses by influencing plant secondary metabolism, detoxifying heavy metals, regulating ethylene levels and emitting volatile organic compounds [47,48]. Similarly, AMF contributes to increased nutrient (N, P, K, Ca and Mg) availability and uptake [49,50]. AMF also augments water absorption by extending the root surface area by mycorrhizal hyphae [51]. Both PGPR and AMF stimulate plant photosynthesis by releasing beneficial phytohormones such as indole-3-acetic acid (IAA), cytokinins and gibberellins, as well as antioxidants, siderophores, enzymes and vitamins, and increasing the stomatal conductance and transpiration rate in plants [50,52,53]. It was demonstrated that biofertilizers,

such as *Bacillus* species, have the potential to increase the production of antioxidants, improve the availability of nutrients such as nitrogen (N), phosphorus (P) and potassium (K) and enhance the growth of hydroponically cultivated lettuce. In the treatments of 50% MF + PGPR, 50% MF + AMF and 50% MF + Microalgae, the biofertilizer statistically increased the firmness of 'Batavia type' lettuce heads by 37.10%, 35.84% and 28.09%, respectively, compared to the 50% MF treatment.

Regarding fresh weight, the application of 100% MF (389.1 g) and 50% MF + PGPR (391.6 g) treatments was found to belong to the same statistical group. The PGPR treatment resulted in a 52.2% increase in lettuce weight compared to the 50% MF treatment. When AMF and microalgae were added to 50% MF, the lettuce weight reached 321.2 g and 312.4 g, respectively (Table 4). Bhat et al. [54] indicated that PGPR, AMF and algae contain biofertilizers that help in increasing crop productivity by way of an increased biological nitrogen fixation, an increased availability or uptake of nutrients or an increased absorption and stimulation of plant growth hormones, antibiosis, by the decomposition of organic residues. Moreover, incorporating biofertilizers to substitute a portion of chemical fertilizers not only diminishes the quantity and expense of chemical fertilizers but also mitigates environmental pollution.

**Table 4.** Effects of the biofertilizers on lettuce plant growth parameters—II.

| Treatments | Lettuce Weight (g plant$^{-1}$) | Leaf Area (cm$^2$ plant$^{-1}$) | Number of Leaves (number plant$^{-1}$) | Dry Matter Ratio in Leaf (%) |
|---|---|---|---|---|
| 100% MF | 389.1 ± 15.7 [a] | 5969 ± 297 [a] | 38.76 ± 1.1 [a] | 4.27 |
| 50% MF | 257.3 ± 13.3 [c] | 4445 ± 394 [c] | 34.30 ± 1.8 [c] | 4.51 |
| 50% MF + Microalgae | 312.4 ± 17.6 [b] | 5064 ± 299 [bc] | 35.90 ± 1.4 [bc] | 4.53 |
| 50% MF + PGPR | 391.6 ± 7.1 [a] | 5882 ± 765 [ab] | 37.10 ± 1.2 [ab] | 4.92 |
| 50% MF + AMF | 321.2 ± 22.4 [b] | 5321 ± 666 [ab] | 36.10 ± 1.8 [bc] | 4.63 |
| LSD$_{0.05}$ | 51.108 | 829.86 | 2.531 | ns |
| *p*-value | 0.0004 | 0.0101 | 0.0278 | 0.100 |

MF: Mineral fertilizer, PGPR: *Plant growth-promoting rhizobacteria*, AMF: Arbuscular mycorrhizal fungi, LSD: The least significant difference between the means ($p < 0.05$), ns: Non-significant. Four biological replicates were employed for each biofertilizer treatment, with each biological replicate comprising five technical replications. Different letters indicate significant differences.

The impact of biofertilizers on the leaf area and leaf number of lettuce plants was statistically significant. PGPR, AMF and microalgae respectively increased the leaf area by 32.33%, 19.71% and 13.73% compared to the 50% MF treatment. The number of leaves per plant in the treatments of 50% MF + B, 50% MF + AMF and 50% MF + microalgae was found to be 8.16%, 5.24% and 4.66% higher compared to the 50% MF treatment. The leaf constitutes the principal photosynthetic apparatus in plants. The increase in the leaf area observed in this context could have been due to the production of plant growth regulators by the microorganism and an enhanced nutrient availability of the biofertilizers. Consequently, this phenomenon led to the promotion of the plant growth and yield. An efficient photosynthetic organ, the increasing leaf area by the biofertilizers likely contributed to the generation of additional plant carbohydrates [31,55]. In the hydroponic cultivation of baby spinach and basil, it was reported that there were increases in leaf area when synthetic mineral fertilizers were reduced by 50% and biofertilizers were employed [23,24]. The rate of dry matter accumulation in lettuce leaves ranged from 4.92% (50% MF + PGPR) to 4.27% (100% MF). While there was no statistically significant difference in terms of dry matter content, it is noteworthy that lettuce cultivated with PGPR has exhibited a higher level of dry matter accumulation (Table 4).

*3.2. Effects of Biofertilizers on Leaf Color Properties*

The luminosity (L), chromatic values (a and b) and SPAD-chlorophyll of the leaves were not significantly influenced by the application of biofertilizers (Table 5). Nevertheless, the 100% MF and 50% MF treatments exhibited the highest and lowest luminosity of the

leaves, with the L values recording 44.04 and 38.06, respectively. Compared to the 50% MF treatment, the biofertilizers increased the leaf luminosity L values by 8.93%, 9.72% and 10.77% in the cases of microalgae, PGPR and AMF, respectively.

**Table 5.** Effects of biofertilizers on leaf color characteristics of lettuce grown in floating hydroponic culture.

| Treatments | L* | a* | b* | SPAD-Chlorophyll |
|---|---|---|---|---|
| 100% MF | 44.04 | −11.16 | 36.29 | 32.82 |
| 50% MF | 38.06 | −10.85 | 31.44 | 30.86 |
| 50% MF + Microalgae | 42.00 | −10.99 | 36.15 | 31.79 |
| 50% MF + PGPR | 42.30 | −11.11 | 37.62 | 32.00 |
| 50% MF + AMF | 42.16 | −10.51 | 34.34 | 31.82 |
| LSD$_{0.05}$ | ns | ns | ns | ns |
| *p*-value | 0.070 | 0.2313 | 0.0536 | 2.446 |

L*: Luminosity, a*: The red/green coordinate, with +a* indicating red and −a* indicating green, b*: The yellow/blue coordinate, with +b* indicating yellow and −b* indicating blue, MF: Mineral fertilizer, PGPR: Plant growth-promoting rhizobacteria, AMF: Arbuscular mycorrhizal fungi, LSD: The least significant difference between the means ($p < 0.05$), ns: Non-significant. Four biological replicates were employed for each biofertilizer treatment, with each biological replicate comprising five technical replications.

Luminosity stands as a key quality characteristic that consumers consider when choosing food, particularly leafy vegetables. The brightness of leaves primarily results from wax on their surfaces, which may be influenced by the environmental conditions during growth [39]. Dasgan and Temtek [56] reported that biofertilizer increased the luminosity; in the case of lettuce grown in a salty soilless medium using cocopeat, the luminosity (L) value increased with the application of biofertilizers; the control recorded 32.93, whereas mycorrhiza, microalgae and bacteria resulted in values of 53.96, 52.20 and 49.21, respectively. Özer Uyar and Mismil [57] demonstrated that applying *Chlorella vulgaris* microalgae led to an increase in the luminosity (L) of mint plants cultivated in a deep-water hydroponic system. In contrast to lettuce, biofertilizers decrease the luminosity (L) in baby spinach cultivated in a floating hydroponic system [40]. The 'a' scale distinguishes between red and green, with a positive value representing red and a negative value indicating green. While no statistically significant difference was observed, the biofertilizers exhibited a stronger inclination toward the green coloration [58]. On the other hand, the 'b' scale discerns between yellow and blue, with a positive value denoting yellow and a negative value denoting blue. Although a slight enhancing effect of biofertilizers on the SPAD-chlorophyll content was observed, no significant difference was found among the applications [8]. Gupta et al. [59] utilized the beneficial fungus *Talaromyces* strain as a biostimulant for lettuce growth, and they reported an increase in photosynthetic pigments.

### 3.3. Effects of Biofertilizers on Lettuce Yield

The differences in the lettuce yield are statistically significant. The highest yield, 16.51 kg m$^{-2}$, was obtained from the 100% MF plants. This is followed by 16.24 kg m$^{-2}$ in the 50% MF + PGPR group, and these two are statistically in the same group. The third-highest yield, 13.87 kg m$^{-2}$, is obtained from the 50% MF + AMF group, while 50% MF + Microgae is in fourth place with 13.08 kg m$^{-2}$, forming the same statistical group as mycorrhiza. The lowest lettuce yield, 11.47 kg m$^{-2}$, was recorded from the 50% MF group (Figure 3). When mineral fertilizers were reduced by 50% and PGPR was added, the lettuce yield per unit area increased by 41.58% compared to the 50% MF treatment. The PGPR application was found to be in the same statistical group as 100% MF and differed only by 1.63%. The applications of 50% MF + AMF and 50% MF + microalgae increased the lettuce yield by 20% and 13.6%, respectively, compared to 50% MF.

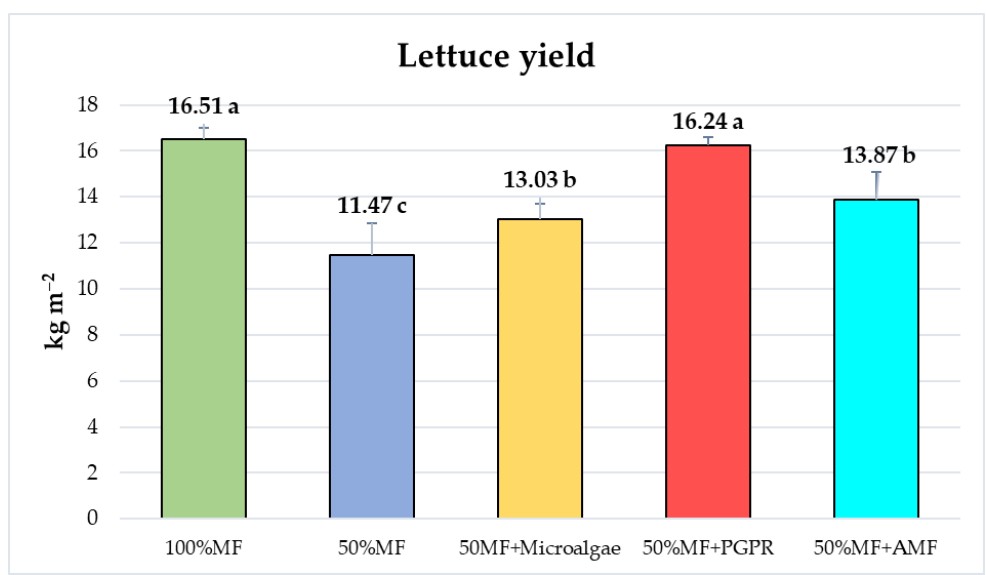

**Figure 3.** Effects of bio-fertilizers on the lettuce yield when mineral fertilizers are reduced by 50% in floating hydroponic culture. MF: Mineral Fertilizer, PGPR: Plant growth-promoting rhizobacteria, AMF: Arbuscular mycorrhizal fungi. Different letters within a column indicate significant differences. Four biological replicates were employed for each biofertilizer treatment, with each biological replicate comprising five technical replications.

In the study conducted by Dasgan et al. [39], it was observed that the hydroponically grown basil leaf yield increased by approximately 18.94% with the application of PGPR, by 13.94% with AMF treatment and by 5.72% with microalgae treatment compared to the 50% MF treatment. The supplementation of biofertilizers increased the leaf weight of baby spinach grown in hydroponic culture by 12%, 8% and 6% in the AMF, PGPR and microalgae treatments, respectively, compared to the 50% MF treatment [40].

It has been reported that biofertilizers positively affect the physiological and biochemical properties such as the leaf growth, biomass, stomatal conductivity, water and nitrogen use efficiency and hormonal activity and ultimately increase the yield and quality of lettuce [52]. Introducing biofertilizers may stimulate the synthesis of biologically active compounds, including phytohormones, amino acids and water-soluble vitamins [27,47]. These phytohormones contribute to vital aspects of plant development and root extension [52]. Studies have indicated that applying biofertilizers exerts a stimulating effect on plants through the influence of hormones, nitrogen fixation, phosphate solubilization and siderophore production. This practice not only reduces the reliance on chemical mineral fertilizers but also enhances the overall plant growth and productivity [30,58,60].

### 3.4. Effects of Biofertilizers on Leaf Nutritional and Antioxidan Compounds

The total soluble solids (TSS) content in leaves exhibited significant differences (Table 6). The application of microalgae and 100% MF led to increases of 27.2% and 29.2%, respectively, compared to the 50% MF application. On the other hand, AMF and PGPR applications resulted in more modest increases of 18.9% and 8%, respectively. The applications did not significantly affect the pH of lettuce leaves, which ranged between 5.99 and 6.09. The electrical conductivity (EC) was measured between 7.71 and 8.52 dS m$^{-1}$ across all applications. Statistically, the highest EC was observed in the 100% MF and microalgae applications. Dasgan et al. [39] stated that the introduction of AMF and PGPR led to heightened carbohydrate production through enhanced photosynthesis and improved nutrient absorption. This, in turn, was associated with a significant increase in both the EC and the total soluble solids in pepper fruits. Ergun et al. [41] reported that reduced mineral fertilizer doses, in combination with *Chlorella vulgaris*, in lettuce plants grown in a floating hydroponic culture, resulted in increased levels of total soluble solids (TSS) and vitamin C

while simultaneously reducing the nitrate content. It has been indicated that TSS and EC in soilless grown tomato fruits are increased by microalgae–*Chlorella vulgaris* [61].

**Table 6.** Effects of bio-fertilizers on the TSS, pH and EC of the lettuce leaf.

| Treatments | TSS (%) | pH | EC (dS m$^{-1}$) |
|---|---|---|---|
| 100% MF | 3.18 ± 0.15 [ab] | 6.09 | 8.52 ± 0.55 [a] |
| 50% MF | 2.50 ± 0.18 [c] | 5.99 | 7.71 ± 0.05 [c] |
| 50% MF + Microalgae | 3.23 ± 0.26 [a] | 6.05 | 8.49 ± 0.51 [ab] |
| 50% MF + PGPR | 2.70 ± 0.30 [c] | 6.04 | 7.91 ± 0.51 [bc] |
| 50% MF + AMF | 2.97 ± 0.43 [b] | 6.05 | 7.83 ± 0.70 [c] |
| LSD$_{0.05}$ | 0.258 | ns | 0.669 |
| *p*-value | 0.0002 | 0.109 | 2.623 |

MF: Mineral Fertilizer, PGPR: *Plant growth-promoting rhizobacteria*, AMF: Arbuscular mycorrhizal fungi, LSD: The least significant difference between the means (*p* < 0.05), ns: Non-significant. Four biological replicates were employed for each biofertilizer treatment, with each biological replicate comprising five technical replications. Different letters indicate significant differences.

Biofertilizers enhance plant growth by producing plant growth regulators, phosphorus solubilization, the availability of mineral elements and biological nitrogen fixation. All of these positive effects may increase the nutritional and antioxidant content of lettuce grown using hydroponic biofertilizers [30].

The cultivation of lettuce plants in a floating culture system has significantly impacted vitamin C production. Specifically, applications of 50% MF combined with biofertilizers yielded statistically higher vitamin C levels than plants grown without biofertilizers. In comparison to the 50% MF application, vitamin C production demonstrated significant increases, namely, 88% with PGPR, 51.2% with AMF and 38.6% with microalgae applications. The utilization of AMF, PGPR and microalgae in cultivating baby spinach and basil plants in a hydroponic system with a 50% reduced mineral fertilizer was reported to enhance the vitamin C content. PGPR has increased the production of vitamin C by more than 40% in hydroponically grown Lollo Rosso lettuce [48]. According to Stojanović et al. [62], the use of microbiological fertilizers containing *Trichoderma* spp. led to a notably increased concentration of vitamin C in lettuce.

The total phenol and flavonoid contents were significantly increased by biofertilizers (Table 7). The total phenolic content exhibited the most significant increase of 27.51% in the AMF application compared to the 50% MF control. All other applications showed a consistent, albeit less pronounced, increase in the phenolic content, ranging from 0.68% to 4.88%. The AMF can stimulate the accumulation of carotenoids, phenolics, anthocyanins, chlorophylls and tocopherols in the leaves of various food crops [63].

**Table 7.** Effects of bio-fertilizers on vitamin C and total phenol and flavonoids.

| Treatments | Vitamin C (mg 100 g FW$^{-1}$) | Total Phenolics (mg GA 100 g FW$^{-1}$) | Total Flavonoids (mg RU 100 g FW$^{-1}$) | Nitrate (mg kg FW$^{-1}$) |
|---|---|---|---|---|
| 100% MF | 7.80 ± 0.96 [c] | 62.45 ± 10.4 [b] | 96.70 ± 3.9 [c] | 462 ± 25 [a] |
| 50% MF | 7.00 ± 0.46 [c] | 61.39 ± 8.0 [b] | 134.81 ± 12.3 [b] | 168 ± 12 [c] |
| 50% MF + Microalgae | 9.70 ± 0.41 [b] | 64.69 ± 1.8 [b] | 139.97 ± 3.4 [b] | 320 ± 18 [b] |
| 50% MF + PGPR | 13.20 ± 0.13 [a] | 61.81 ± 3.0 [b] | 144.90 ± 1.6 [b] | 536 ± 19 [a] |
| 50% MF + AMF | 10.58 ± 1.26 [b] | 78.28 ± 5.2 [a] | 182.37 ± 9.9 [a] | 342 ± 26 [b] |
| LSD$_{0.05}$ | 1.307 | 8.249 | 22.802 | 81.651 |
| *p*-value | <0.0001 | 0.0036 | <0.0001 | <0.0001 |

FW: Lettuce leaves fresh weight, GA: Gallic acid, RU: Rutin, MF: Mineral Fertilizer, PGPR: *Plant growth-promoting rhizobacteria*, AMF: Arbuscular mycorrhizal fungi, LSD: Least significant difference between the means (*p* < 0.05), ns: Non-significant, LSD: The least significant difference between the means (*p* < 0.05). Four biological replicates were employed for each biofertilizer treatment, with each biological replicate comprising five technical replications. Different letters indicate significant differences.

In cultivating baby spinach and basil in hydroponics, the use of AMF and PGPR increased the total phenolic content, respectively [39,40]. The highest amount of flavonoids in lettuce leaves was 182.37 mg RU 100 $g^{-1}$, which was 35.3% higher in the AMF application compared to the 50% MF control. However, it is worth noting that while microalgae and PGPR applications produced higher flavonoid levels, these differences were not statistically significant. In the cultivation of basil in hydroponics, using PGPR increased the total flavonoids [39]. In open-flow hydroponics using the *Bacillus* biofertilizer, lettuce plants exhibited significantly enhanced antioxidant activity, including total phenols and flavonoids [38]. Lettuce inoculated with PGPR has shown significant increases in the content of phenolics and flavonoids. AMF enhances the quality of crops, particularly in terms of the antioxidant capacity, carotenoids, volatile compounds, minerals, vitamins and flavor compounds, as reported by Hart et al. [64]. Gupta et al. [59] utilized the beneficial fungus *Talaromyces* strain as a biostimulant for lettuce growth, and they reported an increase in total phenolics.

### 3.5. Nitrate Concentration in Lettuce Leaves

The nitrate levels were found to be the lowest at 168 mg $kg^{-1}$ in the 50% MF application, while the highest nitrate level of 536 mg $kg^{-1}$ was recorded in the 50% MF + PGPR treatment (Table 7). This may indicate the fixation of atmospheric nitrogen by PGPR. The applications of 50% + Microalgae (320 mg $kg^{-1}$) and 50% + AMF (345 mg $kg^{-1}$) resulted in nitrate levels lower than those in the 100% MF. Kaymak et al. [65] also documented similar findings, showing that a PGPR mixture increased the nitrate content of lettuce leaves. Nevertheless, in contrast to our results, some studies have reported that the application of PGPR reduced the nitrate content in lettuce leaves [66,67]. Gupta et al. [59] utilized the beneficial fungus Talaromyces strain as a biostimulant for lettuce growth, and they reported a decrease in nitrate levels.

Research findings have demonstrated that nitrate levels in curly lettuce can vary widely, ranging from 16 to 3400 mg $kg^{-1}$ FW, with an average of 1601 mg $kg^{-1}$ FW from a dataset of 301 samples [68]. In our investigation, we found that nitrate concentrations remained well below the established thresholds for potential harm to human health. The commercialization threshold, as set by the European Commission (EC Reg. No. 1258/2011), stands at 5000 mg $kg^{-1}$ FW. This threshold applies to protected-grown lettuce cultivated under cover, and our study aligns with this criterion since we conducted our research during the period from October to March [68].

### 3.6. Effects of Biofertilizers on Macro and Micro Nutrients in Lettuce Leaves

All biofertilizers significantly enhanced the concentrations of macro-nutrient elements, namely, nitrogen, phosphorus, potassium, calcium and magnesium [27,58] These increments were even higher than those resulting from a 100% mineral fertilizer application (Table 8).

**Table 8.** Effects of biofertilizers on macro nutrient concentrations in lettuce leaves (%).

| Treatments | N | P | K | Ca | Mg |
|---|---|---|---|---|---|
| 100% MF | 5.37 ± 0.29 [b] | 0.23 | 8.42 ± 0.69 [bc] | 0.83 ± 0.44 [bc] | 1.10 ± 0.03 [bc] |
| 50% MF | 3.65 ± 0.69 [c] | 0.20 | 7.46 ± 0.73 [d] | 0.74 ± 0.35 [c] | 1.00 ± 0.07 [c] |
| 50% MF + Microalgae | 5.24 ± 0.15 [b] | 0.24 | 9.38 ± 0.91 [ab] | 1.06 ± 0.53 [a] | 1.28 ± 0.17 [a] |
| 50% MF + PGPR | 6.46 ± 0.32 [a] | 0.25 | 9.70 ± 0.75 [a] | 0.97 ± 0.79 [ab] | 1.21 ± 0.07 [ab] |
| 50% MF + AMF | 6.12 ± 0.80 [a] | 0.24 | 8.29 ± 0.73 [cd] | 0.96 ± 0.35 [ab] | 1.20 ± 0.07 [ab] |
| LSD$_{0.05}$ | 0.720 | ns | 0.964 | 0.1960 | 0.1564 |
| *p*-value | <0.0001 | 0.0592 | 0.0020 | 0.0275 | 0.0195 |

MF: Mineral Fertilizer, PGPR: *Plant growth-promoting rhizobacteria*, AMF: Arbuscular mycorrhizal fungi, LSD: The least significant difference between the means ($p < 0.05$), ns: Non-significant. Four biological replicates were employed for each biofertilizer treatment, with each biological replicate comprising five technical replications. Different letters indicate significant differences.

The nitrogen (N) content in leaf samples ranged from 3.65% in the 50% MF-control group to 6.46% in the group treated with PGPR. Statistical analysis revealed that both PGPR and AMF applications resulted in significantly higher N concentrations. Moreover, leaf phosphorus (P) levels varied from 0.20% in the 50% MF application to 0.25% in the PGPR-treated group. The PGPR application led to a 25% increase in phosphorus compared to the 50% MF control. The potassium (K) content in the leaves was statistically highest, measuring 9.7% in the PGPR-treated group and 9.38% in the microalgal-treated group. The lowest content was observed at 7.46% in the 50% MF control application. The calcium (Ca) concentrations ranged from 0.74% in the 50% MF group to 1.06% in the 50% MF combined with microalgae application. It is worth noting that the experimental lettuce plants exhibited a minor increase in the magnesium (Mg) content compared to the 50% MF-control. The highest magnesium content, measuring 1.28%, was observed in the microalgal-treated group.

Mycorrhizal symbiosis improved the mineral status of the lettuce plants. It has been reported that the presence of AMF in association with host plants can lead to a twofold increase in nutrient concentrations compared to their non-mycorrhizal counterparts [69]. This increase in the nutrient concentration is directly attributed to the expanded hyphal network of AMF within the root zone, making essential nutrients readily available to the host plants through absorption and transportation. Additionally, AMF can enhance water use efficiency, which, in turn, can potentially improve overall plant nutrition [63].

PGPR employs several direct and indirect mechanisms to regulate nutrient flow, thereby promoting plant growth. Some of these mechanisms include the production of phytohormones (auxin, cytokinin, gibberellin and kinetin), nitrogen fixation, the solubilization of organic and inorganic minerals and the synthesis of bio-control agents such as siderophores, hydrogen cyanide, antibiotics and enzymes [58,70,71]. Bhat et al. [54] stated that nitrogen-fixing bacteria and blue green algae convert atmospheric nitrogen to organic compounds that plants use.

The iron (Fe) concentration in lettuce leaves ranged from 69.23 ppm to 101.42 ppm. Statistically, the highest Fe concentration was observed at 101.42 ppm in the PGPR-treated group (Table 9). The manganese (Mn) content ranged from 21.52 ppm in the 50% MF-control to 33.38 ppm in the 50% MF + Microalgae group. All bio-fertilizers increased in Mn content, with the highest Mn concentrations observed in the microalgae and PGPR-treated groups. The zinc (Zn) analysis in lettuce leaves yielded results ranging from 51 ppm in the 50% MF control to 70 ppm in the 50% MF + PGPR group. The bio-fertilizers significantly increased the concentration of zinc (Zn), especially the application of PGPR, which showed a remarkable increase. Additionally, the PGPR biofertilizer significantly elevated copper (Cu) levels compared to the control application, while other bio-fertilizers exhibited similar effects. The copper (Cu) concentrations ranged from 3.99 ppm in the 50% MF + AMF group to 5.99 ppm in the 100% MF group.

**Table 9.** Effects of biofertilizers on micro nutrient concentrations in lettuce leaves (mg kg$^{-1}$).

| Treatments | Fe | Mn | Zn | Cu |
|---|---|---|---|---|
| 100% MF | 78.77 ± 6.84 [b] | 26.87 ± 2.33 [bc] | 65.80 ± 8.50 [ab] | 5.99 ± 0.58 [a] |
| 50% MF | 69.23 ± 1.28 [b] | 21.52 ± 3.35 [c] | 51.00 ± 3.63 [c] | 4.29 ± 0.61 [bc] |
| 50% MF + Microalgae | 74.29 ± 3.31 [b] | 33.38 ± 4.48 [a] | 55.44 ± 5.63 [bc] | 4.72 ± 0.38 [b] |
| 50% MF + PGPR | 101.42 ± 7.53 [a] | 31.52 ± 5.30 [ab] | 70.94 ± 6.97 [a] | 5.71 ± 0.40 [a] |
| 50% MF + AMF | 80.71 ± 8.64 [b] | 26.03 ± 3.04 [bc] | 56.12 ± 2.29 [bc] | 3.99 ± 0.06 [c] |
| LSD$_{0.05}$ | 12.91 | 6.083 | 11.921 | 0.667 |
| *p*-value | 0.0016 | 0.0084 | 0.0188 | <0.0001 |

MF: Mineral Fertilizer, PGPR: *Plant growth-promoting rhizobacteria*, AMF: Arbuscular mycorrhizal fungi, LSD: The least significant difference between the means ($p < 0.05$). Four biological replicates were employed for each biofertilizer treatment, with each biological replicate comprising five technical replications. Different letters indicate significant differences.

Mycorrhizal plants can increase the uptake of metal nutrients such as Cu, Zn and Fe through their root colonization and the extension of extraradical hyphae. These hyphae offer a significantly larger surface area than roots alone, reducing the diffusion distance and consequently facilitating the absorption of immobile metal nutrients [72]. Rana et al. [73] reported that PGPR inoculation led to an enhancement in the Zn, Fe and Cu contents of wheat plants, indicating a possible role in improving the translocation of micronutrients. Karlidag et al. [74] found that PGPR inoculations increased the Ca, K, Fe, Cu, Mn and Zn levels in leaves. This increase may also be explained by the production of organic acids by both plants and bacteria in the rhizosphere, which lowers the pH and stimulates the availability of nutrients. Pacheco et al. [75] demonstrated that a combined seaweed-bacteria fertilizer enriched lettuce leaf micronutrient contents, specifically zinc and manganese.

*3.7. Heat Map and Principal Component Analysis*

The data for thirty parameters, including the plant growth parameters, lettuce yield, quality traits, antioxidants, nitrate content and elemental concentrations of lettuce leaves, were visualized using a heatmap (Figure 4). The higher lettuce yield, lettuce weight, lettuce height, leaf area, stem diameter, leaf dry matter, lettuce firmness, lettuce circumference and vitamin C, N, P, K, Fe, Mn, Zn and Cu concentrations were obtained through the application of PGPR. The highest total phenolics and flavonoids were achieved using AMF, while the highest TSS (Total Soluble Solids) content was obtained when microalgae were applied. Heat maps offer a visual means of comprehending numerical data, allowing for a comprehensive view of multiple data points and their interrelationships using a color-coded system. Figure 4 presents a consolidated overview of the biofertilizers' effectiveness regarding lettuce plant growth, yield and quality parameters. Heat maps have applications beyond data visualization; they can be employed more literally, such as to delineate 'hot and cold' regions (e.g., represented by red and blue color schemes) on a map. In the context of this study, the '50% MF + PGPR' application displayed the most intense red and light blue hues, whereas the '50% MF' application exhibited predominantly dark to medium blue shades.

The Principal Component Analysis (PCA), conducted on thirty variables, reveals that AMF and microalgae exhibit a closer proximity in the scatterplot. In contrast, treatments involving 50% mineral fertilizer (50% MF) and 100% mineral fertilizer (100% MF) demonstrate a distinct separation and dissociation from biofertilizer applications. Notably, the PGPR treatment forms a discernible grouping in the scatterplot, indicating a unique response pattern in relation to the other treatments.

This study was conducted under natural sunlight in a greenhouse using a floating hydroponic culture. In future research, the use of biofertilizers in indoor vertical farming systems with different LED lightings, under different photoperiods, and their recyclable application in the different hydroponic systems, such as ebb–flow, aeroponcis and NFT, could be investigated. While new biofertilizers, such as vermicompost, can be explored, novel PGPR, AMF and microalgae species can be examined within existing biofertilizers.

Metabolomics has emerged as a valuable technology for elucidating a comprehensive biochemical profile of a biological system [76]. For future studies, it will be beneficial to identify economically significant and biologically functional phytochemicals in hydroponically cultivated lettuce nourished with biofertilizers.

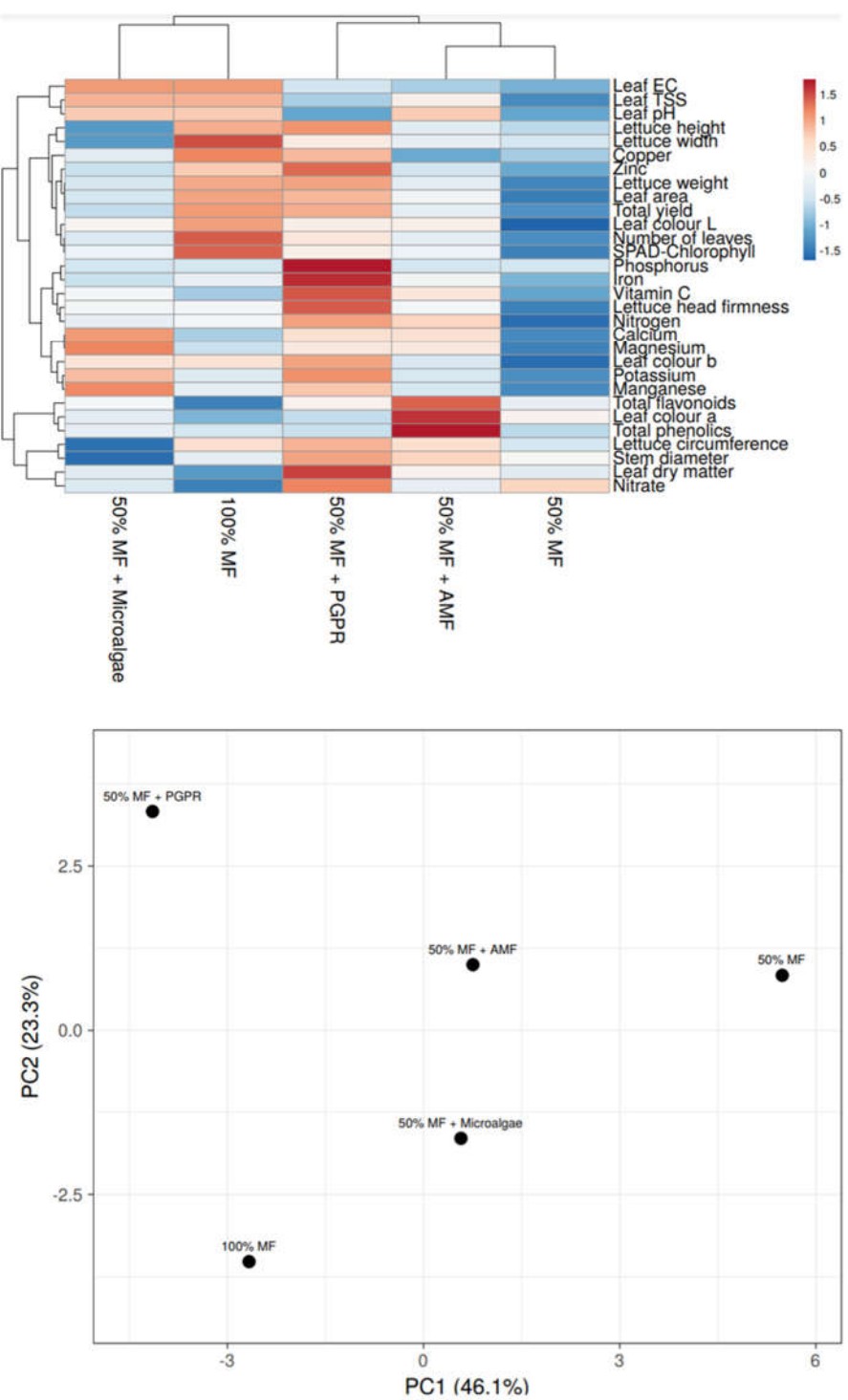

**Figure 4.** Heatmap and Principal Component Analysis (PCA) illustrating the relationship between biofertilizers and variables.

## 4. Conclusions

Combining plant growth-promoting rhizobacteria (PGPR) with a 50% mineral fertilizer regimen has shown promising results in achieving lettuce yields comparable to those achieved with a 100% mineral fertilizer application. This approach boosts plant growth and improves the antioxidant and mineral content of lettuce leaves. Therefore, based on the findings of this study, we recommend the incorporation of PGPR for hydroponic lettuce cultivation. Utilizing PGPR as bio-inoculants can improve nutrient availability, reducing



the reliance on chemical fertilizers. Transitioning to biofertilizers as a partial substitute for chemical fertilizers not only reduces the quantity and cost of chemical fertilizers but also helps mitigate the environmental pollution resulting from their extensive use. This shift towards reduced chemical fertilizer usage aligns with promoting sustainable agricultural practices and represents a significant step in advancing environmentally friendly and economically viable lettuce production.

**Author Contributions:** All the authors contributed to this research. H.Y.D. and D.Y. designed the experiment. Conceptualization; data curation; formal analysis; investigation; resources; funding acquisition: H.Y.D., D.Y., K.Z. and B.I. Supervision; writing—review and editing: H.Y.D. and N.S.G. All authors have read and agreed to the published version of the manuscript.

**Funding:** This work was supported by the Cukurova University Research Foundation (BAP) under project number FYL-2018-11173.

**Data Availability Statement:** The data presented in this study are available in the article.

**Conflicts of Interest:** The authors declare no conflict of interest.

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
