# Peer review of "Enhancing the Yield, Quality and Antioxidant Content of Lettuce through Innovative and Eco-Friendly Biofertilizer Practices in Hydroponics"

_horticulturae, doi:10.3390/horticulturae9121274_

Round 1

Reviewer 1 Report

Comments and Suggestions for Authors

The manuscript is well written and has scientific merit. However, I would like to suggest some corrections for the betterment of the manuscript.

1. Lines 11 and 42: Please complete the statement to express a full understanding and clarify the background of the research.

2. Lines 35 and 41: Please provide citations.

3. Line 75: "Soil problem." Please replace or modify the word to continue the idea.

4. Line 80: Please delete or move this sentence to keep the continuation of the statement.

5. Line 98: Please modify the word "furthermore.".

6. Line 120: Please superscript the value.

7. Figure 1: Please provide scale measurements for the photograph.

8. Tables: Please explain the alphabet in the table (what are they representing?) footnote.

9. Please use ISO-guided standard abbreviations to express the measurement units.

Comments on the Quality of English Language

Minor spell checks for typos are required, like in lines 305, 565.

Author Response

I have uploaded the responses as a  of the Reviewer-1 as PDF file.

Reviewer 2 Report

Comments and Suggestions for Authors

This study thoroughly explored the impacts of different biofertilizers on lettuce seedling growth and quality within a hydroponic cultivation system, reducing reliance on mineral fertilizers. The obtained results highlighted the advantageous effects of PGPR on plant yield and quality, offering a pragmatic approach to sustaining productivity while minimizing chemical fertilizer usage. The manuscript's overall presentation is well-structured and composed. My primary concern is the lack of confirmation regarding whether the utilized microbes genuinely colonized the rhizosphere or roots in this hydroponic system. Additionally, the manuscript could benefit from these minor revisions:

1. Did the authors alter the cultivation solutions during the 40-day treatment?

  1. 2. In lines 238-239, could the authors elaborate on the biological significance of "head firmness"?
  2. 3. Ensure standard errors for each plant trait are provided in all tables.
  3. 4. Lines 431 and Table 4: Based on Table 4, the lowest nitrate content appears under the 100%MF treatment, not 50%MF.
  4. 5. How were the figures conducted, specifically the creation of the phylogenetic relationship between different plant traits? Additionally, where is the output for the principal component analysis?
Comments on the Quality of English Language

Fine

Author Response

I have uploaded the responses of the Reviewer-2 as PDF file.

Reviewer 3 Report

Comments and Suggestions for Authors

Regarding this manuscript, I provide the following comments:

  1. I recommend citing more articles published in 2023 or with the year of publication in 2024 in the introduction and throughout the manuscript to provide fresher information to the readers;

  2. I suggest adding a high-resolution figure (like a graphical abstract) summarizing the main findings and to make the manuscript more illustrative;

  3. In the discussion, I recommend providing perspectives on future studies based on the findings of the current manuscript;

  4. For future studies, the authors could suggest large-scale metabolomic analyses. By the way, metabolomics could help to evaluate other non-enzymatic antioxidants.

  5. Do the authors intend to perform quantifications of antioxidant enzymes?

  6. In the abstract, when the authors mention antioxidants, they should explain which antioxidants they are talking about

  7. Methods: How many technical replicates and biological replicates were used for statistical analyses?

  8. During my reading, I found several errors in English or writing style.

For example, the following sentence sounds grammatically incorrect to me and I did not understand the full meaning of the sentence: “ Furthermore, biofertilizers as to replace part of the use of chemical fertilizers reduces amount and cost of chemical fertilizers and thus prevents the environment pollution from extensive application of these fertilizers”.

This is just one example. There are several others.

Please revise the entire manuscript to avoid these errors. If necessary, ask a native speaker or an English expert to review each sentence.

  1. The abstract should mention more methodological information

  2. The authors mentioned that the experiment was carried out in 2019. Did the authors also repeat the experiment recently? How many times was the experiment repeated to ensure the quality of the results?

  3. The explanation about the use of technical replicates and biological replicates for statistical analyses should be cited in the captions of figures and tables of the results.

  4. I suggest significantly improving the discussion by comparing the results of the present research to studies published more recently on the same research topic.

    13. Please add a abbreviation list if possible.

Comments on the Quality of English Language

Please see my comments to the authors.

Author Response

I have uploaded the responses of the Reviewer-3 as PDF file.

Round 2

Reviewer 3 Report

Comments and Suggestions for Authors

I have no more comments.